# Increased Job Burnout and Reduced Job Satisfaction for Nurses Compared to Other Healthcare Workers after the COVID-19 Pandemic

Petros Galanis [1,*], Ioannis Moisoglou [2], Aglaia Katsiroumpa [1], Irene Vraka [3], Olga Siskou [4], Olympia Konstantakopoulou [5], Evangelia Meimeti [6] and Daphne Kaitelidou [5]

1 Clinical Epidemiology Laboratory, Faculty of Nursing, National and Kapodistrian University of Athens, P.C. 11527 Athens, Greece; aglaiakat@nurs.uoa.gr
2 Faculty of Nursing, University of Thessaly, P.C. 41500 Larissa, Greece; iomoysoglou@uth.gr
3 Department of Radiology, P. & A. Kyriakou Children's Hospital, P.C. 11527 Athens, Greece; irenevraka@yahoo.gr
4 Department of Tourism Studies, University of Piraeus, P.C. 18534 Piraeus, Greece; olsiskou@nurs.uoa.gr
5 Center for Health Services Management and Evaluation, Faculty of Nursing, National and Kapodistrian University of Athens, P.C. 11527 Athens, Greece; olympiak1982@hotmail.com (O.K.); dkaitelid@nurs.uoa.gr (D.K.)
6 3rd Regional Health Authority of Macedonia, P.C. 54623 Thessaloniki, Greece; e.meimeti@gmail.com
* Correspondence: pegalan@nurs.uoa.gr

**Abstract:** Nurses experience high levels of job burnout and low levels of job satisfaction, while the COVID-19 pandemic has deteriorated working conditions. In this context, our aim was to compare levels of job burnout and job satisfaction among nurses and other healthcare workers (HCWs) after the COVID-19 pandemic. Moreover, we investigated the influence of demographics and job characteristics on burnout and satisfaction. We conducted a cross-sectional study with 1760 HCWs during June 2023. We used the single-item burnout measure and the "Job Satisfaction Survey". In our sample, 91.1% of nurses experienced high levels of burnout, while the respective percentage for the other HCWs was 79.9%. Nurses' satisfaction was lower than other HCWs. In particular, 61.0% of nurses experienced low levels of satisfaction, while the respective percentage for the other HCWs was 38.8%. Multivariable analysis identified that nurses, HCWs with an MSc/PhD diploma, shift workers, and those who considered their workplace as understaffed had higher burnout score and lower satisfaction score. Our results showed that the nursing profession was an independent factor of burnout and satisfaction. Several other demographic and job characteristics affected burnout and satisfaction. Policy makers, organizations, and managers should adopt appropriate interventions to improve work conditions.

**Keywords:** nurses; healthcare workers; burnout; satisfaction; workplace

## 1. Introduction

Globally, nurses are the largest professional group in healthcare [1]. They spend most of their working time providing direct or indirect care to patients [2] and the quality of patient care depends largely on them and their work environment [3]. Healthcare organizations worldwide face significant challenges. These include an aging population and an increase in chronic diseases [4], the need for continuous improvement in the quality and safety of services [5], and, at the same time, the limitation of available resources [6]. To this already demanding working environment, the pandemic of COVID-19 was added, where, for at least two years, healthcare professionals and especially nurses found themselves working in extremely difficult conditions and were called upon to provide care in urgent and very serious cases. In all of these challenges, nurses are called upon to play a leading role in addressing them. There are two important variables that are predictors of nurses'

efforts to meet these challenges: job satisfaction and job wellbeing, and in particular the absence of burnout.

Nurses' satisfaction helps to create a climate of safety and improves the quality and safety of nursing care [7,8]. Also, patients who are hospitalized in wards where nurses are satisfied report satisfaction with the nursing care provided [9]. When healthcare organizations ensure nurses' job satisfaction, then the likelihood of turnover intention and quitting their existing employment are reduced [9,10]. Additionally, turnover intention among nurses increased during and after the COVID-19 pandemic [11]. Through nurses' satisfaction, a healthy working environment is created, where satisfaction from communication with colleagues and supervisors increases and job performance is promoted [12,13]. Good communication with colleagues and supervisors is a prerequisite for providing quality and safe care and also helps to support each other.

Despite the important multidimensional role of job satisfaction, the managements of healthcare organizations have failed to ensure high levels of job satisfaction among nurses. Studies at different levels of healthcare delivery showed that nurses either report moderate satisfaction or almost half of them report dissatisfaction [14,15]. The main factors associated with nurses' dissatisfaction are shift work, high workload, low payment, leadership style, lack of attention to the social status of the nursing profession, and lack of autonomy [14,16–18]. These factors are almost exclusively related to the working environment of nurses and, therefore, the management of healthcare organizations, in their efforts to increase nurses' satisfaction, should take them into account. The low satisfaction of health professionals is a persistently negative situation and should be among the priorities of the administrations in terms of human-resources management.

A condition that has a high prevalence in nursing staff is burnout. In the pre-COVID-19 period, the overall pooled prevalence of burnout symptoms among global nurses was 11.23% [19], while other studies put the percentage of exhausted nurses at 30% [20,21]. As in the case of satisfaction, the environment in which nurses provide their services is associated with an increased likelihood of burnout syndrome. Better collegial nurse–physician relationships, increased nurse participation in hospital affairs, nurse-manager ability, leadership, support of nurses, low/inadequate nurse staffing levels, ≥12-h shifts, and low autonomy have been identified as the main work-related factors associated with the occurrence of burnout [22–24]. The intense exposure of health professionals to very serious COVID-19 incidents, intubations, and deaths also affected the mental health of health professionals. Under these demanding conditions, with a lack of resources and support, a significant proportion of nurses experienced moral distress, depression, anxiety, and burnout, and were at high risk for having post-traumatic stress disorder (PTSD). In fact, burnout and moral distress were both moderately to highly correlated with anxiety, depression, and PTSD symptoms [25]. The high demands of working with COVID-19 patients, combined with the weaknesses of organizational support, made nurses feel exhausted and often unable to cope with their work and personal responsibilities [26].

The impact of burnout is multifaceted and influences the nurses themselves (organizational commitment and performance) together with patients (quality of care, safety, and satisfaction) [27]. The combination of the negative effects of burnout with the high prevalence rates among nurses makes managing burnout imperative to create a healthy workplace and, at the same time, a safe care environment.

During the period of the COVID-19 pandemic, health systems were under great pressure as patient admissions (both in general wards and ICUs) increased significantly. The severity of the conditions of patients with COVID-19, combined with the special care conditions (use of personal protective equipment and risk of infection) created a high workload and work intensity. Also, nurses experienced extremely difficult working conditions, such as longer working time in quarantine areas, working in a high-risk environment, in hospitals with inadequate and insufficient material and human resources, increased workloads, and lower levels of specialized training regarding COVID-19 [28]. These factors greatly increased both nurses' levels of burnout and job dissatisfaction during the COVID-19 pandemic, with satisfaction having a significant negative correlation with burnout [28–32].

Healthcare organizations have now returned to their pre-COVID-19 functioning. However, to the best of our knowledge, there are no studies that investigated healthcare workers' job burnout or job satisfaction after the COVID-19 pandemic. Thus, the main aim of our study was to compare levels of job burnout and job satisfaction among nurses and other healthcare workers after the COVID-19 pandemic. Moreover, we investigated the influence of demographic and job characteristics on healthcare workers' job burnout and satisfaction.

## 2. Materials and Methods

### 2.1. Study Design

A cross-sectional study was conducted in Greece. We collected data during June 2023. We invited healthcare workers to participate in our study through several sources: face-to-face interviews, healthcare workers' groups in social media, and e-mail contacts. In particular, we created an online version of our study questionnaire using Google Forms. Then, we posted the online questionnaire on healthcare workers' groups on social media, namely Facebook and LinkedIn. Moreover, we sent the online questionnaire to our e-mail contacts asking healthcare workers to participate in our study. Finally, we approached healthcare workers through a face-to-face strategy. In that case, we had a printed version of the questionnaire and we asked participants to fill it out. Also, we asked them if they want to complete the online version of the questionnaire instead of the printed one. In case of a positive answer, we sent them an e-mail with the online questionnaire. We obtained a convenience sample in our study. The response rate was unknown since we cannot estimate the number of healthcare workers that had knowledge of our study through social media or e-mails but they did not complete the questionnaire.

We applied the following inclusion criteria: (a) healthcare workers in clinical settings including hospitals, health centers, primary healthcare services, etc., (b) healthcare workers who have been working during the COVID-19 pandemic (2020–2023), (c) healthcare workers from any profession, i.e., nurses, physicians, midwives, psychologists, pharmacists, etc. We excluded healthcare workers that have not been working during the pandemic in clinical settings. The number of healthcare workers working in clinical settings in Greece is about 60,000.

### 2.2. Measures

We measured the following demographic characteristics of the participants: gender (females or males), age (continuous variable), and educational level (MSc/PhD diploma or not).

Moreover, we measured the following job characteristics: years of clinical experience (continuous variable), shift work (no or yes), full-time job (no or yes), job sector (public or private), and understaffed workplace (no or yes).

We used the Greek version of the single-item burnout measure (SIB) to measure levels of burnout among our healthcare workers [33]. In particular, Greek scholars validated the SIB in a sample of healthcare workers, i.e., 963 nurses. Thus, the study population in their study is similar to our study population. Moreover, they investigated the reliability of the SIB by performing a test-retest study and measuring the intraclass correlation coefficient between the two measurements. Also, they found that the concurrent and discriminant validities of the SIB were excellent. In that case, they found a high correlation between the SIB and three other valid instruments, namely the Copenhagen Burnout Inventory, the Patient Health Questionnaire-4, and the COVID-19 burnout scale. The SIB measure is a brief, valid, and reliable tool to measure burnout [34]. Hansen and Pit validated the SIB in a sample of general practitioners working in clinical settings in Australia. In particular, they found very good concurrent validity of the SIB since it was significantly associated with three other valid instruments, i.e., the Maslach Burnout Inventory, Kessler Psychological Distress Scale, and 36-Item Short Form Survey. Also, they found that the SIB had high sensitivity (79%) and specificity (87%) using the Maslach Burnout Inventory as the standard. The SIB includes only the following question "In a scale from 0 (not at all) to 10 (totally),

how tired do you feel?". Answers are on a continuous scale from 0 to 10. Higher values indicate higher levels of burnout. We considered values from 0 to 3 to indicate a low level of burnout, values from 4 to 6 to indicate a moderate level of burnout, and values from 8 to 10 to indicate a high level of burnout.

We used the "Job Satisfaction Survey" (JSS) to measure levels of satisfaction in our sample [35]. The JSS has 36 items and answers are on a six-point Likert scale: disagree very much (1); disagree moderately (2); disagree slightly (3); agree slightly (4); agree moderately (5); agree very much (6). Some items are reverse-scored. Thus, the JSS takes values from 36 to 216, with higher values indicative of higher levels of satisfaction. According to the JSS score, individuals are classified as those with low satisfaction, moderate satisfaction, and high satisfaction.

### 2.3. Ethical Considerations

The Ethics Committee of the Faculty of Nursing, National and Kapodistrian University of Athens approved our study protocol (approval number; 451, June 2023). Moreover, we applied the guidelines of the Declaration of Helsinki. We informed healthcare workers of the aim and the design of our study. Healthcare workers could withdraw from the study at any time. Healthcare workers gave their informed consent to participate. We included the question "Do you agree to participate in this study?" in the online version of the questionnaire. This question was the first one and it was obligatory. Only a positive answer gave the opportunity to healthcare workers to continue with other questions, while a negative answer did not allow them to participate in our study. A final approval button should be checked by the participants to submit their answers to Google Forms. Thus, healthcare workers can give up at any point of the online questionnaire by just closing the web page. In that case, their answers have not been submitted to Google Forms.

### 2.4. Statistical Analysis

We use numbers and percentages to present categorical variables and mean (standard deviation, SD), minimum value, and maximum value to present continuous variables. The Kolmogorov–Smirnov test showed that age, score on the single-item burnout measure, and score on the "Job Satisfaction Survey" followed a normal distribution, while years of clinical experience did not follow a normal distribution. Demographic and job characteristics were the independent variables, while job burnout and job satisfaction were the dependent variables. We used the chi-square trend test to identify differences between nurses and other healthcare workers and levels of job burnout and job satisfaction. Moreover, we used the Spearman's correlation coefficient to estimate the correlation between age and years of clinical experience. Since the correlation between age and years of clinical experience was very high ($r = 0.9$, $p < 0.001$), we used only years of clinical experience in the regression analysis. First, we conducted a univariate linear regression analysis and, then, we constructed a final multivariable linear regression model to eliminate confounding. In this context, we identified the independent effect of demographic and job characteristics on job burnout and job satisfaction among healthcare workers. We calculated unadjusted and adjusted coefficient betas, 95% confidence intervals (CI), and *p*-values. Moreover, we calculated the adjusted coefficient of determination for the final multivariable models. The *p*-values less than 0.05 were considered statistically significant. We used IBM SPSS 21.0 (IBM Corp. Released 2012. IBM SPSS Statistics for Windows, Version 21.0. Armonk, NY, USA: IBM Corp.) for the analysis.

## 3. Results

### 3.1. Healthcare Workers

The study population included 1760 healthcare workers. Detailed demographic and job characteristics of the healthcare workers are shown in Table 1. In our sample, 53.8% were nurses and 46.2% were other healthcare workers. The mean age was 41.1 years (SD = 9.8). The minimum age was 25 years and the maximum was 67 years. The majority of healthcare

workers was females (80.2%) and had a full-time job (96.7%). Among our healthcare workers, 60.6% possessed an MSc/PhD diploma. More than half of the participants were shift workers (55.5%), while almost three out of four (77.5%) have been working in the public sector. The mean year of clinical experience was 16.2 (SD = 9.5). The minimum clinical experience was 3 years and the maximum was 40 years. Most of the healthcare workers (83.3%) reported that they have been working in understaffed workplaces.

**Table 1.** Demographic and job characteristics of healthcare workers.

| Characteristics | N | % |
|---|---|---|
| Gender | | |
| Females | 1412 | 80.2 |
| Males | 348 | 19.8 |
| Age [a] | 41.1 | 9.8 |
| MSc/PhD diploma | | |
| No | 694 | 39.4 |
| Yes | 1066 | 60.6 |
| Job | | |
| Nurses | 946 | 53.8 |
| Other healthcare workers | 814 | 46.2 |
| Shift work | | |
| No | 784 | 44.5 |
| Yes | 976 | 55.5 |
| Full-time job | | |
| No | 58 | 3.3 |
| Yes | 1702 | 96.7 |
| Sector | | |
| Private | 394 | 22.4 |
| Public | 1366 | 77.6 |
| Understaffed workplace | | |
| No | 294 | 16.7 |
| Yes | 1466 | 83.3 |
| Years of clinical experience [a] | 16.2 | 9.5 |

[a] mean, standard deviation.

### 3.2. Study Scales

The mean burnout score was 6.9 (SD = 2.6), with a median value of 8, and a range from 0 to 10. Nurses experienced moderate and high levels of burnout more often than other healthcare workers ($p < 0.001$). In particular, 91.1% of nurses experienced moderate/high levels of burnout, while the respective percentage for the other healthcare workers was 79.9% (Table 2). Additionally, 8.9% of nurses experienced low levels of burnout, while the respective percentage for the other healthcare workers was 20.1%. In total, 69.1% of our sample experienced high levels of burnout, 16.8 experienced moderate levels, and 14.1% experienced low levels.

The mean satisfaction score was 108.2 (SD = 30.9) with a median value of 106 and a range from 36 to 216. Nurses' satisfaction was lower than other healthcare workers ($p < 0.001$). In particular, 61.0% of nurses experienced low levels of satisfaction, while the respective percentage for the other healthcare workers was 38.8% (Table 2). Moreover, 39.0% of nurses experienced moderate/high levels of satisfaction, while the respective percentage for the other healthcare workers was 61.2%. Among our total sample, 50.7% experienced low levels of satisfaction, 36.1% experienced moderate levels, and 13.1% experienced high levels.

**Table 2.** Levels of job burnout and job satisfaction among healthcare workers.

| | Levels of Job Burnout | | | | | | *p*-Value |
|---|---|---|---|---|---|---|---|
| | Low | | Moderate | | High | | |
| | N | % | N | % | N | % | |
| Nurses | 84 | 8.9 | 192 | 20.3 | 670 | 70.8 | <0.001 |
| Other healthcare workers | 164 | 20.1 | 104 | 12.8 | 546 | 67.1 | |
| Total | 248 | 14.1 | 296 | 16.8 | 1216 | 69.1 | |
| | Levels of Job Satisfaction | | | | | | |
| | Low | | Moderate | | High | | |
| | N | % | N | % | N | % | |
| Nurses | 577 | 61.0 | 288 | 30.4 | 81 | 8.6 | <0.001 |
| Other healthcare workers | 316 | 38.8 | 348 | 42.8 | 150 | 18.4 | |
| Total | 893 | 50.7 | 636 | 36.1 | 231 | 13.1 | |

*3.3. Regression Analysis*

Univariate linear regression analysis revealed that gender ($p < 0.001$), job profession ($p < 0.001$), educational level ($p = 0.01$), shift work ($p < 0.001$), full-time job ($p < 0.001$), job sector ($p < 0.001$), understaffed workplace ($p < 0.001$), and clinical experience ($p < 0.001$) were associated with the job-burnout score. Then, we created a multivariable model to eliminate confounding factors. Multivariable linear regression analysis identified that females ($p < 0.001$), nurses ($p = 0.005$), and healthcare workers with an MSc/PhD diploma ($p = 0.004$) had higher job-burnout scores. Moreover, levels of burnout were higher among shift workers ($p < 0.001$), those with a full-time job ($p = 0.017$), and those that have been working in the private sector ($p = 0.002$). Additionally, healthcare workers that assessed their workplace as understaffed were more burnt out ($p < 0.001$). We found a positive relationship between clinical experience and job burnout ($p < 0.001$). The proportion of variance in the job-burnout score, which was explained by the independent variables, was 17.4%. Detailed linear regression results with the job-burnout score as the dependent variable are shown in Table 3.

**Table 3.** Linear regression analysis with job-burnout score as the dependent variable.

| | Univariate Model | | Multivariable Model | |
|---|---|---|---|---|
| **Independent Variables** | **Unadjusted Coefficient Beta (95% CI)** | ***p*-Value** | **Adjusted Coefficient Beta (95% CI)** | ***p*-Value** |
| Males vs. females | −0.82 (−1.13 to −0.51) | <0.001 | −0.65 (−0.94 to −0.37) | <0.001 |
| Nurses vs. other healthcare workers | 0.59 (0.35 to 0.84) | <0.001 | 0.34 (0.10 to 0.57) | 0.005 |
| MSc/PhD diploma | 0.33 (0.08 to 0.58) | 0.010 | 0.35 (0.11 to 0.58) | 0.004 |
| Shift work | 1.57 (1.34 to 1.81) | <0.001 | 1.38 (1.14 to 1.63) | <0.001 |
| Full-time job | 1.33 (0.64 to 2.02) | <0.001 | 0.78 (0.14 to 1.42) | 0.017 |
| Public-sector job | 0.67 (0.37 το 0.96) | <0.001 | −0.47 (−0.77 to −0.17) | 0.002 |
| Understaffed workplace | 1.93 (1.62 to 2.25) | <0.001 | 1.32 (0.99 to 1.64) | <0.001 |
| Years of clinical experience | 0.05 (0.03 to 0.06) | <0.001 | 0.05 (0.04 to 0.07) | <0.001 |

CI: confidence interval; Adjusted $R^2$ for the model = 17.4%; *p*-value for ANOVA < 0.001.

The linear regression analysis with job-satisfaction score as the dependent variable is shown in Table 4. First, we conducted a univariate linear regression analysis and we found that gender ($p = 0.001$), job profession ($p < 0.001$), educational level ($p < 0.001$), shift work ($p < 0.001$), full-time job ($p = 0.008$), job sector ($p < 0.001$), understaffed workplace ($p < 0.001$), and clinical experience ($p = 0.002$) were associated with job-satisfaction score. Then, we constructed a final multivariable linear regression model. In that case, we found that nurses ($p < 0.001$), healthcare workers with an MSc/PhD diploma ($p = 0.003$), and shift workers were less satisfied with their job. Additionally, job satisfaction was lower among

healthcare workers in the public sector ($p = 0.002$), and among those who considered their workplace as understaffed ($p < 0.001$). Moreover, we found that clinical experience was associated with decreased job satisfaction ($p = 0.009$). The proportion of variance in the job-satisfaction score, which was explained by the independent variables, was 18.9%.

**Table 4.** Linear regression analysis with job-satisfaction score as the dependent variable.

| Independent Variables | Univariate Model | | Multivariable Model | |
|---|---|---|---|---|
| | Unadjusted Coefficient Beta (95% CI) | *p*-Value | Adjusted Coefficient Beta (95% CI) | *p*-Value |
| Males vs. females | 6.31 (2.70 to 9.92) | 0.001 | 2.34 (−0.98 to 5.67) | 0.167 |
| Nurses vs. other healthcare workers | −15.12 (−17.93 to −12.32) | <0.001 | −11.49 (−14.22 to −8.76) | <0.001 |
| MSc/PhD diploma | −5.84 (−8.79 to −2.90) | <0.001 | −4.24 (−6.99 to −1.49) | 0.003 |
| Shift work | −16.94 (−19.74 to −14.15) | <0.001 | −11.32 (−14.15 to −8.49) | <0.001 |
| Full-time job | −10.94 (−19.02 to −2.87) | 0.008 | −2.70 (−10.11 to 4.71) | 0.475 |
| Public-sector job | −16.01 (−19.39 to −12.62) | <0.001 | −5.62 (−9.11 to −2.14) | 0.002 |
| Understaffed workplace | −26.00 (−29.67 to −22.32) | <0.001 | −17.74 (−21.51 to −13.97) | <0.001 |
| Years of clinical experience | −0.24 (−0.40 to −0.09) | 0.002 | −0.20 (−0.35 to −0.05) | 0.009 |

CI: confidence interval; Adjusted $R^2$ for the model = 18.9%; *p*-value for ANOVA < 0.001.

## 4. Discussion

This study was conducted in June 2023. As the COVID-19 pandemic has subsided, protection measures have been suspended and the operation of healthcare organizations has returned to the pre-COVID-19 era. To the best of our knowledge, this is the first study that assessed job burnout and job satisfaction in healthcare workers after the COVID-19 pandemic. Therefore, as there are no similar studies in the literature conducted in the post-COVID period, our findings will be discussed and interpreted according to those of studies conducted during the COVID-19 pandemic.

According to the study findings, nurses experienced very high rates of burnout and dissatisfaction, higher compared to other healthcare professionals. These findings are consistent with similar studies in the literature during the pandemic period, where nurses have been found to experience higher rates of burnout and dissatisfaction compared to other healthcare professionals [36–38]. The finding that is particularly significant and needs interpretation is that levels of burnout and dissatisfaction are higher now than in the midst of the pandemic, although the impact of the pandemic on healthcare systems has subsided. Therefore, while the workload and work intensity due to the pandemic have eased, some factors affecting nurses' satisfaction and burnout seem to remain. Two possible interpretations can be given for this finding. The first is that nurses, as frontline professionals, experienced intense psychological distress due to fear of being infected with COVID-19, transmission of the virus to their family, fear of death from COVID-19, lack of knowledge of how to use personal protective equipment or lack of equipment, and lack of support. All these burdened nurses' mental health and affected their level of satisfaction and burnout [39–41]. It is very likely that the influence of the above factors continues to affect nurses. The second possible interpretation relates to the organization and operation of healthcare services. Before the pandemic occurred, major issues such as understaffing, overtime work, the inability of leadership to support staff, and increased workloads negatively affected nursing staff and undermined the quality of care. Obviously, during the pandemic period, it was not possible for healthcare organizations' managers to resolve these issues, which continue to this day to plague the functioning of healthcare organizations and certainly negatively affect nurses' satisfaction and burnout. As health organizations were under great pressure due to the increased number of patients with COVID-19, they reduced the operation of some of their services, such as primary care, psychiatry, orthopedics, cardiology, and some scheduled surgeries. Immediately after the pandemic subsided, the healthcare organizations resumed operations in the pre-COVID-19 period with full operation of all services. Consequently, there was no rest period for the

nurses and so, exhausted as they were, they continued to provide their services in an equally demanding work environment [42].

The present study highlighted understaffing as an organizational factor associated with burnout and job dissatisfaction. This finding is consistent with the results of an extensive study at 256 hospitals in the United States shortly before the pandemic outbreak. The study highlighted the nursing understaffing of these hospitals as well as the impact of understaffing on burnout, satisfaction together with the intention to leave the job of nursing staff, and the quality of care provided [43]. Understaffing feeds the intention to leave a job and, at the same time, the intention to leave a job exacerbates understaffing. Essentially, on the one hand, nurses are caught in a vicious cycle and, on the other hand, the organization is spending money and other resources to recruit and train new staff. Although, until now we knew about nurses quitting the profession due to job dissatisfaction or burnout [44], however, during the pandemic, a particularly worrying phenomenon called "quiet quitting" began to develop rapidly. In the case of quiet quitting, the employees do not actually quit their jobs but they continue to work with a much lower performance. Also, they are doing the bare minimum, perhaps a step short of actively trying to get fired, while not actively going above and beyond [45,46]. A recent study showed that a large percentage of nurses are quiet quitters and they choose quiet quitting more often than other healthcare professionals [47]. Nurses' job burnout is a positive predictor of quiet quitting, while job satisfaction is a negative one of quiet quitting [48]. Due to the pandemic, finding a job is difficult. Many job vacancies have been lost and, therefore, nurses resigning and changing career orientation is not easy, so they preferred to stay in the nursing profession. Although by keeping nurses in the profession, the understaffing of hospitals does not worsen; however, this is one side of the coin. The quiet quitting of them undermines the efficiency of the healthcare organization and compromises the quality and safety of the care provided, as there is a risk that nursing tasks are left unfinished and patients' needs are not met. Rudimentary nursing work and effort, due to quiet quitting, is very likely to be the "Achilles' heel" of the health services in their attempt to achieve their mission.

According to our study, educational level and work experience were found to be associated with the occurrence of job dissatisfaction and burnout. These findings are in contrast to similar ones in the literature, as high educational level and many years of work experience are predictors of job satisfaction and reduced burnout [49,50]. Regarding the educational level, this finding can be interpreted in the context of the functioning of the healthcare system in Greece. The majority of healthcare organizations are public. In order to obtain a position of responsibility (nurse manager), there is relevant legislation that sets out the qualifications required and the points an employee receives for each qualification. Among the qualifications that receive high points is the possession of a master's or doctoral degree. Consequently, the possession of a Master's/PhD degree gives an advantage during the evaluation process assignment of a position of responsibility. However, for the last 15 years, no assessments have been carried out in accordance with legislation. It is very likely that postgraduate degree holders would have had expectations of taking up positions of responsibility. Studies in Greece have shown that healthcare professionals are dissatisfied with promotions in their workplace [15,51]. This frustration may be reflected in the results of the present study. Regarding the burnout and dissatisfaction of those with many years of service, this is probably related to staff fatigue. In particular, over the last 20 years, Greece has been at the bottom of the OECD countries in terms of the ratio of nurses per 1000 inhabitants (3.4 nurses per 1000 inhabitants) [52], when the average number of OECD countries is 8.8. This long-standing shortage of nursing staff, with its consequences (workload, overtime, shift work, etc.), can be reflected in the findings of the present study.

Our study also has some limitations. First, we used a convenience sample that is not representative of the population of nurses in Greece. For example, most of our healthcare workers were females and possessed an MSc/PhD diploma. Therefore, we cannot generalize our findings. Further studies in different countries, cultures, and clinical settings could add valuable information. Furthermore, comparison studies between different countries

will give us the opportunity to infer valid conclusions. Second, we used the single-item burnout measure to measure levels of burnout. Although the SIB measure is a valid and reliable tool, there are also other ones that measure job burnout, such as the Maslach Burnout Inventory and the Copenhagen Burnout Inventory. These tools are more comprehensive, including subfactors such as detachment, personal burnout, client-related burnout, etc. However, we chose the SIB measure in our study in order to improve the response rate since other similar tools comprise too many questions. Third, we used self-reported instruments to collect our data. Therefore, information bias was probable in our study. Fourth, we measured the impact of several demographic and job characteristics on burnout and satisfaction. However, several other predictors can influence burnout and satisfaction, e.g., salary, managerial position, organizational culture, resilience, social support, etc.

## 5. Conclusions

This study highlights the high rates of job dissatisfaction and burnout among healthcare workers in the post-COVID-19 era. In fact, these rates are even higher than those of the pandemic period, when health systems were under extreme pressure. Understanding the factors that affect nurses' burnout and satisfaction is essential to improve their daily caring interventions [53]. The characteristics of the working environment of healthcare professionals, which negatively affect their job satisfaction and wellbeing, appear to be long-standing weaknesses of health systems. Long-standing organizational problems and inefficiencies in health systems remain unresolved up to now. As a result, they continue to negatively affect healthcare professionals, leading them to burnout and dissatisfaction. Healthcare professionals are the most important resource in a health-service organization, as the optimal management of other resources and the outcomes of the organization depend to a large extent on them. Therefore, the extremely significant consequences of burnout and dissatisfaction, combined with the ongoing challenges to health systems, the latest being the pandemic, make the resolution of organizational issues fundamental to ensure effectiveness, efficiency, and the delivery of quality health services.

**Author Contributions:** Conceptualization, P.G.; methodology, P.G., I.M. and D.K.; software, P.G. and O.K.; validation, A.K., I.V., O.S., O.K. and E.M.; formal analysis, P.G., A.K. and I.V.; investigation, P.G., A.K., I.V., O.S., O.K. and E.M.; resources, P.G., A.K., I.V., O.S., O.K. and E.M.; data curation, I.M., A.K., I.V., O.S., O.K. and E.M.; writing—original draft preparation, P.G., I.M., A.K. and I.V.; writing—review and editing, P.G., I.M., A.K., I.V., O.S., O.K., E.M. and D.K.; supervision, P.G.; project administration, P.G. and D.K. All authors have read and agreed to the published version of the manuscript.

**Funding:** This research received no external funding.

**Institutional Review Board Statement:** The study was conducted in accordance with the Declaration of Helsinki and approved by The Ethics Committee of the Faculty of Nursing, National and Kapodistrian University of Athens (approval number; 451, June 2023).

**Informed Consent Statement:** Informed consent was obtained from all subjects involved in the study.

**Data Availability Statement:** The data presented in this study are available on request from the corresponding author.

**Public Involvement Statement:** There was no public involvement in any aspect of this research.

**Guidelines and Standards Statement:** This manuscript was drafted against the (STROBE) for a cross-sectional study, descriptive research.

**Acknowledgments:** We acknowledge all the participants who make this study possible.

**Conflicts of Interest:** The authors declare no conflict of interest.

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
