# Peer review of "Increased Job Burnout and Reduced Job Satisfaction for Nurses Compared to Other Healthcare Workers after the COVID-19 Pandemic"

_nursrep, doi:10.3390/nursrep13030095_

Round 1

Reviewer 1 Report

I have worked extensively with nurses, HCWs and doctors during and in-between the different COVID-peaks. This article is well-written, the used instruments are reliable. Perhaps I would have like some more information about the Greek translation of the burnout assessment scale which has been used by the researchers/authors.

I would have liked some more peri-COVD and post-COVID contextual factors that may have contributed to the negative mood and burnout of the nurses and HCWs. There may have been trauma-related factors and moral injuries, they remain under-reported and have not been identified by the research team. 

Other institutional factors may have contributed to the burnout feelings of the medical staff i.e. in the wake of COVID, many medical treatments had to be re-started (such as non-urgent surgery, cancer treatment, psychiatric treatment, etc.), nurses were left with no time for recovery, no decompression after COVID, which may have contributed to the general feeling of abandonment and tiredness.

Next to references to other studies, contextual references and more info about lived experiences of nurses and HCWs, would have increased the ecological quality of this manuscript.

If any adaptations in this may could be carried out, that would be fine.

Author Response

Dear Reviewer,

Thank you very much for the peer review of the manuscript “Increased job burnout and reduced job satisfaction for nurses compared to other health care workers after the COVID-19 pandemic”. Thank you for your comments, which have improved the quality of the manuscript. We have addressed all the comments (highlighted in yellow) in the revised text.

Please, find below an item-by-item answer to your comments. Hoping the revised manuscript fulfils the journal’s standards, we thank you for your courtesy.

We are looking forward to your response.

Yours sincerely,

Petros Galanis, Assistant Professor

Comment:

I have worked extensively with nurses, HCWs and doctors during and in-between the different COVID-peaks. This article is well-written, the used instruments are reliable. Perhaps I would have like some more information about the Greek translation of the burnout assessment scale which has been used by the researchers/authors.

Answer: Done

Dear Reviewer we add the following text in the Methods section (2.2. Measures).

We used the Greek version of the single item burnout measure (SIB) to measure levels of burnout among our health care workers [33]. In particular, Greek scholars validated the SIB in a sample of health care workers, i.e. 963 nurses. Thus, study population in their study is similar to our study population. Moreover, they investigated the reliability of the SIB performing a test-retest study and measuring intraclass correlation coefficient between the two measurements. Also, they found that the concurrent and discriminant validity of the SIB were excellent. In that case, they found a high correlation between the SIB and three other valid instruments, namely the Copenhagen Burnout Inventory, the Patient Health Questionnaire-4 and the COVID-19 burnout scale.

Comment:

I would have liked some more peri-COVD and post-COVID contextual factors that may have contributed to the negative mood and burnout of the nurses and HCWs. There may have been trauma-related factors and moral injuries, they remain under-reported and have not been identified by the research team.

Answer: Done

Dear Reviewer,

Regarding the post-COVID era, to the best of our knowledge, there are still no published studies on burnout and satisfaction among health professionals and nurses in particular.

With regard to the COVID era, we added text and two more references in the introduction section on trauma-related factors, which have contributed to the burnout feelings.

We add the following text in Introduction section (lines 76-84):

The intense exposure of health professionals to very serious COVID-19 incidents, intubations and deaths also affected the mental health of health professionals. Under these demanding conditions, with a lack of resources and support, a significant proportion of nurses experienced moral distress, depression, anxiety, burnout and were at high risk for having posttraumatic stress disorder (PTSD). In fact, burnout and moral distress were both moderately to highly correlated with anxiety, depression, and PTSD symptoms [25]. The high demands of working with COVID-19 patients, combined with the weaknesses of organizational support, made nurses feel exhausted and often unable to cope with their work and personal responsibilities [26].

Comment:

Other institutional factors may have contributed to the burnout feelings of the medical staff i.e. in the wake of COVID, many medical treatments had to be re-started (such as non-urgent surgery, cancer treatment, psychiatric treatment, etc.), nurses were left with no time for recovery, no decompression after COVID, which may have contributed to the general feeling of abandonment and tiredness.

Answer: Done

Dear Reviewer,

You rightly write that the restarting of health care organizations’ does not give nurses any room for rest and therefore, exhausted, they continue to offer their services in an equally demanding working environment.

We add the following text in Discussion section (lines 292-299):

As the health organizations were under great pressure due to the increased number of patients with COVID-19, they reduced the operation of some of their services such as primary care, psychiatry, orthopedics, cardiology and some scheduled surgeries. Immediately after the pandemic subsided, the health care organizations resumed operations in the pre-COVID-19 period with full operation of all services. Consequently, there was no rest period for the nurses and so, exhausted as they were, they continued to provide their services in an equally demanding work environment.

Comment:

Next to references to other studies, contextual references and more info about lived experiences of nurses and HCWs, would have increased the ecological quality of this manuscript.

If any adaptations in this may could be carried out, that would be fine.

Answer: Done

Dear Reviewer,

In the Introduction section, where we added the new text (lines 76-84), one reference (26) is from a qualitative study presenting the impact of the pandemic on nurses' work and personal lives.

We add the following text in Introduction section (lines 76-84):

The intense exposure of health professionals to very serious COVID-19 incidents, intubations and deaths also affected the mental health of health professionals. Under these demanding conditions, with a lack of resources and support, a significant proportion of nurses experienced moral distress, depression, anxiety, burnout and were at high risk for having posttraumatic stress disorder (PTSD). In fact, burnout and moral distress were both moderately to highly correlated with anxiety, depression, and PTSD symptoms [25]. The high demands of working with COVID-19 patients, combined with the weaknesses of organizational support, made nurses feel exhausted and often unable to cope with their work and personal responsibilities [26].

Reviewer 2 Report

The research is interesting and worthwhile. 

The only concern I have is the use of a single item variable. You supported the use of the single-item variable but it lacks the support to convince the reader of the soundness of using the single-item variable. The document will be stronger if you explain more from  Galanis et al. and Hansen's works that makes a stronger argument for the use of the single-item variable.

Author Response

Dear Reviewer,

Thank you very much for the peer review of the manuscript “Increased job burnout and reduced job satisfaction for nurses compared to other health care workers after the COVID-19 pandemic”. Thank you for your comments, which have improved the quality of the manuscript. We have addressed all the comments (highlighted in yellow) in the revised text.

Please, find below an item-by-item answer to your comments. Hoping the revised manuscript fulfils the journal’s standards, we thank you for your courtesy.

We are looking forward to your response.

Yours sincerely,

Petros Galanis, Assistant Professor

The only concern I have is the use of a single item variable. You supported the use of the single-item variable but it lacks the support to convince the reader of the soundness of using the single-item variable. The document will be stronger if you explain more from Galanis et al. and Hansen's works that makes a stronger argument for the use of the single-item variable.

Answer: Done

Dear Reviewer we add the following text in the Methods section (2.2. Measures).

We used the Greek version of the single item burnout measure (SIB) to measure levels of burnout among our health care workers [33]. In particular, Greek scholars validated the SIB in a sample of health care workers, i.e. 963 nurses. Thus, study population in their study is similar to our study population. Moreover, they investigated the reliability of the SIB performing a test-retest study and measuring intraclass correlation coefficient between the two measurements. Also, they found that the concurrent and discriminant validity of the SIB were excellent. In that case, they found a high correlation between the SIB and three other valid instruments, namely the Copenhagen Burnout Inventory, the Patient Health Questionnaire-4 and the COVID-19 burnout scale. The SIB measure is a brief, valid and reliable tool to measure burnout [34]. Hansen & Pit validated the SIB in a sample of general practitioners working in clinical settings in Australia. In particular, they found very good concurrent validity of the SIB since it was significantly associated with three other valid instruments, i.e. Maslach Burnout Inventory, Kessler Psychological Distress Scale, and 36-Item Short Form Survey. Also, they found that the SIB had high sensitivity (79%) and specificity (87%) using the Maslach Burnout Inventory as the standard.

Reviewer 3 Report

Dear authors:

I reviewed the manuscript entitled “Increased job burnout and reduced job satisfaction for nurses compared to other health care workers after the COVID-19 pandemic”. Your study aim to compare levels of job burnout and job satisfaction among nurses and other health care workers (HCWs) after the COVID-19 pandemic. It was conducted a cross-sectional study with 1760 HCWs during 20 June 2023. The results showed that nursing profession was an independent factor of burnout and satisfaction. Several other demographic and job characteristics affected burnout and satisfaction. Policy makers, organizations and managers should adopt appropriate interventions to improve work conditions.

The thematic is very interesting and should be a concern to nurse management.

On a few notes to further develop and reflection to improve the manuscript, I would suggest:

1) It isn´t clear if you used an online form of the questionnaire (and what software do you used t create it) or in paper form? But if was in paper form, how the health care works had access to fill the questionnaire, when they have access by social media? (How to deal with distance?)

2) How many health workers do you have in Greece?

3) If you used an online version, in that case how they consent to participate? and how you guarantee the possibility to participants give up in any period of your investigation? (e.g. after fulfill the questionnaire? They have some code that allow the investigators to remove the data?)

4) In 2.4. statistical analysis you should refer the tests you used (e.g. Correlation test- spearman? Pearson?)

5) I suggest to use a point in material and methods to explain more clear the sample characteristics; inclusion and exclusion criterium (did you had in attention if health professional worked directly with patients with covid-19?)

I have nothing to add and I wish you all the best with its publication.

Best regards!

Author Response

Dear Reviewer,

Thank you very much for the peer review of the manuscript “Increased job burnout and reduced job satisfaction for nurses compared to other health care workers after the COVID-19 pandemic”. Thank you for your comments, which have improved the quality of the manuscript. We have addressed all the comments (highlighted in yellow) in the revised text.

Please, find below an item-by-item answer to your comments. Hoping the revised manuscript fulfils the journal’s standards, we thank you for your courtesy.

We are looking forward to your response.

Yours sincerely,

Petros Galanis, Assistant Professor

I reviewed the manuscript entitled “Increased job burnout and reduced job satisfaction for nurses compared to other health care workers after the COVID-19 pandemic”. Your study aim to compare levels of job burnout and job satisfaction among nurses and other health care workers (HCWs) after the COVID-19 pandemic. It was conducted a cross-sectional study with 1760 HCWs during 20 June 2023. The results showed that nursing profession was an independent factor of burnout and satisfaction. Several other demographic and job characteristics affected burnout and satisfaction. Policy makers, organizations and managers should adopt appropriate interventions to improve work conditions.

The thematic is very interesting and should be a concern to nurse management.

On a few notes to further develop and reflection to improve the manuscript, I would suggest:

1) It isn´t clear if you used an online form of the questionnaire (and what software do you used t create it) or in paper form? But if was in paper form, how the health care works had access to fill the questionnaire, when they have access by social media? (How to deal with distance?)

Answer: Done

Dear Reviewer we add the following text in the Methods section (2.1. Study design).

In particular, we created an on-line version of our study questionnaire using Google forms. Then, we posted the on-line questionnaire on health care workers groups in social media, namely Facebook and LinkedIn. Moreover, we sent the on-line questionnaire to our e-mail contacts asking from health care workers to participate in our study. Finally, we approached health care workers through a face-to-face strategy. In that case, we had a printed version of the questionnaire and we asked from participants to fill it. Also, we asked them if they want to complete the on-line version of the questionnaire instead of the printed one. In case of a positive answer we sent to them an e-mail with the on-line questionnaire.

2) How many health workers do you have in Greece?

Answer: Done

The number of health care workers working in clinical settings in Greece is about 60,000.

We add the above information in the Methods section (2.1. Study design).

3) If you used an online version, in that case how they consent to participate? and how you guarantee the possibility to participants give up in any period of your investigation? (e.g. after fulfill the questionnaire? They have some code that allow the investigators to remove the data?)

Answer: Done

Dear Reviewer we add the following text in the Methods section (2.3. Ethical considerations).

We included the question “Do you agree to participate in this study?” in the on-line version of the questionnaire. This question was the first one and it was obligatory. Only a positive answer gave the opportunity to health care workers to continue with other questions, while a negative answer did not allow them to participate in our study. A final approval button should be checked by the participants to submit their answers to Google forms. Thus, health care workers can give up in any period of the on-line questionnaire by just closing the web page. In that case, their answers have not been submitted to Google forms.

4) In 2.4. statistical analysis you should refer the tests you used (e.g. Correlation test- spearman? Pearson?)

Answer: Done

Dear Reviewer we add the following text in the Methods section (2.4. statistical analysis).

We used the chi-square trend test to identify differences between nurses and other health care workers and levels of job burnout and job satisfaction. Moreover, we used the Spearman’s correlation coefficient to estimate the correlation between age and years of clinical experience.

5) I suggest to use a point in material and methods to explain more clear the sample characteristics; inclusion and exclusion criterium (did you had in attention if health professional worked directly with patients with covid-19?)

Answer: Done

Dear Reviewer we add the following text in the Methods section (2.1. Study design).

We applied the following inclusion criteria: (a) health care workers in clinical settings including hospitals, health centers, primary healthcare services etc., (b) health care workers who have been working in clinical settings during the COVID-19 pandemic, (c) health care workers from any profession, i.e., nurses, physicians, midwives, psychologists, pharmacists, etc. We excluded health care workers that have not been working during the pandemic in clinical settings.

We did not include only healthcare workers that have been working directly with COVID-19 patients since the COVID-19 pandemic affects healthcare workers’ burnout and satisfaction in all clinical settings. Thus, we decided to include healthcare workers that have been working during the pandemic in all clinical settings.

I have nothing to add and I wish you all the best with its publication.

Round 2

Reviewer 3 Report

Dear Authors

I believe the manuscript is now more clear. I don´t have anything, more, to suggest to review.

Best regards.